# What's the relative humidity in tropical caves?

**Luis Mejía-Ortíz[1], Mary C. Christman[2], Tanja Pipan[3,4], David C. Culver[5]***

**1** División de Desarrollo Sustentable, Lab. de Bioespeleología y Carcinología, Universidad de Quintana Roo, Cozumel, Mexico, **2** Departments of Biology and Statistics, University of Florida and MCC Statistical Consulting LLC, Gainesville, Florida, United States of America, **3** ZRC SAZU Karst Research Institute, Ljubljana, Slovenia, **4** UNESCO Chair on Karst Education, University of Nova Gorica, Vipava, Slovenia, **5** Department of Environmental Science, American University, Washington, DC, United States of America

* dculver@american.edu

**Data Availability Statement:** All relevant data are within the manuscript and its Supporting Information files.

**Funding:** LMMO was supported by CONACYT (www.conacyt.gob.mx) grant 258494 (Fondo Sectorial de Investigacion para la Educacion). The

## Abstract

Relative humidity (RH) was measured at hourly intervals for approximately one year in two caves at seven stations near Playa del Carmen in Quintana Roo, Mexico. Sistema Muévelo Rico is a 1.1 km long cave with 12 entrances and almost no dark zone. Río Secreto (Tuch) is a large river cave with more than 40 km of passages, and an extensive dark zone. Given the need for cave specialists to adapt to saturated humidity, presumably by cuticular thinning, the major stress of RH would be its deviation from saturation. RH in Río Secreto (Tuch) was invariant at three sites and displayed short deviations from 100% RH at the other four sites. These deviations were concentrated at the end of the nortes and beginning of the rainy season. Three of the sites in Sistema Muévelo Rico showed a similar pattern although the timing of the deviations from 100% RH was somewhat displaced. Four sites in Sistema Muévelo Rico were more variable, and were analyzed using a measure of amount of time of deviation from 100% RH for each 24 hour period. Strong seasonality was evident but, remarkably, periods of constant high humidity were not the same at all sites. In most Sistema Muévelo Rico sites, there was a detectable 24 hour cycle in RH, although it was quite weak in about half of them. For Río Secreto (Tuch) only one site showed any sign of a 24 hour cycle. The troglomorphic fauna was more or less uniformly spread throughout the caves and did not concentrate in any one area or set of RH conditions. Compared to temperature, RH is much more constant, perhaps even more constant than the amount of light. However, changes in RH as a result of global warming may have a major negative effect on the subterranean fauna.

## Introduction

The transition from a surface habitat to the subterranean habitat of caves is a profound one, both physically and biologically. At least three physical attributes of caves change significantly. Light disappears, or at least nearly so—there is at least the theoretical possibility of light in a cave. Badino [1] showed Cerenkov radiation emitted in air, water and rock by cosmic ray muons resulted in light production in caves, and in at least some large chambers, should be detectable by the human eye. Temperature variation is greatly reduced, and hovers around the

funders had no role in study design, data collection and analysis, decision to publish, or preparation of the manuscript. MCC Statistical Consulting LLC provided support in the form of salaries for MCC, but did not have any additional role in the study design, data collection and analysis, decision to publish, or preparation of the manuscript. The specific role of the author is articulated in the author contribution section.

**Competing interests:** MCC Statistical Consulting LLC provided support in the form of salaries for CC. The commercial affiliation of MCC to MCC Statistical Consulting LLC does not alter our adherence to PLOS ONE policies on sharing data and materials.

mean annual surface temperature [2,3]. Relative humidity, the focus of this contribution, does not behave as does temperature, but rather increases relative to surface habitats, and in many cases, remains near saturation [3,4], well above mean annual relative humidity on the surface.

A cave entrance is a transition zone (ecotone) between the illuminated surface environment and the constant darkness of the subsurface. The transition between light and dark is not necessarily abrupt and nearly all caves have a twilight zone of reduced, but not absent, light. In some exceptional cases, some light is present throughout the cave, even when the cave is more than 1000 m in length [1]. Beyond the twilight zone is a zone of fluctuating temperature [2], followed by a zone of constant temperature [3]. The zone of fluctuating temperature may also be extensive and it is not entirely clear whether most caves do in fact have a constant temperature zone [4].

Of these three physical factors, relative humidity has received the least attention. Culver and Pipan [5] argue that light is the only driver of convergent natural selection in subterranean habitats. The biological importance of cave temperature is less obvious but even small differences in temperature, on the order of 1°C, can have a major impact on micro-distribution of cave spiders, and an important feature of niche differentiation among competing species [6,7]. Relative humidity is perhaps the most constant of the three, and Howarth [4,8] argues that it is of profound importance as a selective factor, and that the ability of a terrestrial organisms to survive in an atmosphere of 100% RH requires major morphological changes, especially cuticular thinning. Nicolosi et al. [9] show that the cave isopod *Armadillium lagrecai* was very sensitive to fluctuations in temperature and relative humidity. Howarth [8] showed that longevity for the Hawaiian spider *Lycosa howarthi* was reduced by 25% as a result of a RH drop to only 90% from saturation. Hadley et al. [10] further showed that the cuticle of cave spiders was thinned, compared to surface relatives. This reduction allows individuals to survive water saturated environments, but at a cost of desiccation when relative humidity drops below saturation [8,9]. As they point out, at 100% RH, the subterranean terrestrial environment has some features of an aquatic environment and is certainly an extreme environment.

Each of these three factors is mediated through cave entrances and has a spatial distribution of values that in the deep cave is invariant, or at least assumed to be invariant. For temperature, the presumed invariant value is the mean annual surface temperature [3,11], for light it is its absence. The length of the cave until the invariant zone is reached depends on the particular geometry of the cave, particularly with respect to the size and aspect of the entrance(s) [12]. The invariant zone for relative humidity is saturation (or more properly vapor equilibrium pressure [11]). Since relative humidity has a strong dependence on temperature (the Clausius-Clapeyron equation [13]), relative humidity might be expected to behave in a similar manner to temperature. However, the presence of standing water in many caves obviates this dependency, and relative humidity is typically at saturation [14]

In a previous study [15], we analyzed both spatial and temporal variation in temperature in three caves in Quintana Roo, Mexico, and found a remarkable amount of cyclical variation (both daily and annual) in temperature, both in photic and aphotic zones. The daily signal in some aphotic sites was extremely weak, but the annual cycle had an amplitude of at least 2°C. Cropley [16] reported a minimum of 4°C variation in sites 1800 m from the entrance in two large cave systems in West Virginia. Likewise, Šebela and Turk [17] found 1°C of variation at sites deep in the Postojna Planina Cave System of Slovenia. There may well be zones of constant temperature in these large caves, but the zone of variable temperature is extensive.

Relative humidity in caves, the focus of this study, has been little investigated, either theoretically or empirically, in comparison. There is a rich literature on cave temperatures, including topics such as mean temperature prediction using passage size, entrance size and exterior temperature [18]; time lags between exterior and cave temperature [19,20]; the relationship

between ventilation and temperature [21]; the likely effects of global warming on cave temperature [13], as well as general analytical treatments [11,19].

In an analytical sense, relative humidity is less interesting than temperature in part because it is so dependent on temperature—the vapor equilibrium pressure depends on the temperature of the system and the vapor equilibrium pressure ($g/m^3$) is numerically close to the temperature in ˚C [11]. However, humidity may be generally at saturation (equilibrium) when in close contact of water surfaces and air [11,14]. However, RH is interesting biologically because of the apparent sensitivity of cave organisms to even small deviations from saturation [4,8,10]. It is also of geochemical interest because it is the switch point between evaporation and condensation. For example, carbonate dissolution can occur via condensation corrosion in saturated air [22,23].

Our goals in this study are:

1. Characterize the relative humidity regime, based on hourly samples taken over a year, for a series of sites in two tropical caves.

2. Identify zones of constant humidity, and to characterize short term deviations from saturated relative humidity in these two caves.

3. Detect daily and seasonal cycles in the two caves.

4. Compare humidity patterns with those of temperature and light.

5. Highlight risks of RH changes to the cave fauna as a result of climate change

## Materials and methods

### The study caves

The two caves (Sistema Muévelo Rico and Río Secreto) are located in the Quintana Roo in the Yucatan Peninsula (Fig 1) in an area with one of the highest cave densities of cave passages (mostly flooded) in the world [24,25]. Air filled caves are also numerous and they are constrained to a relatively thin layer of flat-bedded limestone with a depth of 5 to 10 m to the water table, and a surface topography of gentle ridges and swales with an overall relief of 1–5 m [24,26] The area has an annual cycle of precipitation characterized by three seasons: nortes (cold front season between November and February), dry season (March to May), and rainy season (June to October) which is the hurricane season [27]. During the rainy season 70% of the precipitation occurs. The annual mean air temperature is 25.8˚ C and the overall precipitation at Playa del Carmen averages 1500 mm [27].

Sistema Muévelo Rico (20˚32'05.1"N, 87˚12'16.5"W) is located near the settlement of Paamul, in the Mexican state of Quintana Roo (Fig 1). Its surveyed length is 1151 m with a vertical extent of only 4 m [25]. Sistema Muévelo Rico has a large number of entrances, more than 12, if skylights are included. Because of the close proximity of the water table to the surface, vertical development and subterranean terrestrial habitats are very restricted. The cave, with an elevation of 7 m at the entrance, is less than 2 km from the Caribbean Sea. There were seven monitoring points in the cave. It was originally chosen for study because of its extensive twilight zone and extremely small aphotic zone [28].

Río Secreto (20˚35'27"N, 87˚8'3"W) is a shallow, horizontally developed cave with 42 km of surveyed passages (Fig 1). It is a tourist cave and the tours are conducted in a small section of the cave. The main entrance is 5 km from the Caribbean coast and 12 km NE of Sistema Muévelo Rico. Tides can affect the water table in Río Secreto up to several cm [27]. There were seven monitoring points clustered in the vicinity of the Tuch entrance (Fig 1), and we refer to the cave as Río Secreto (Tuch) throughout.

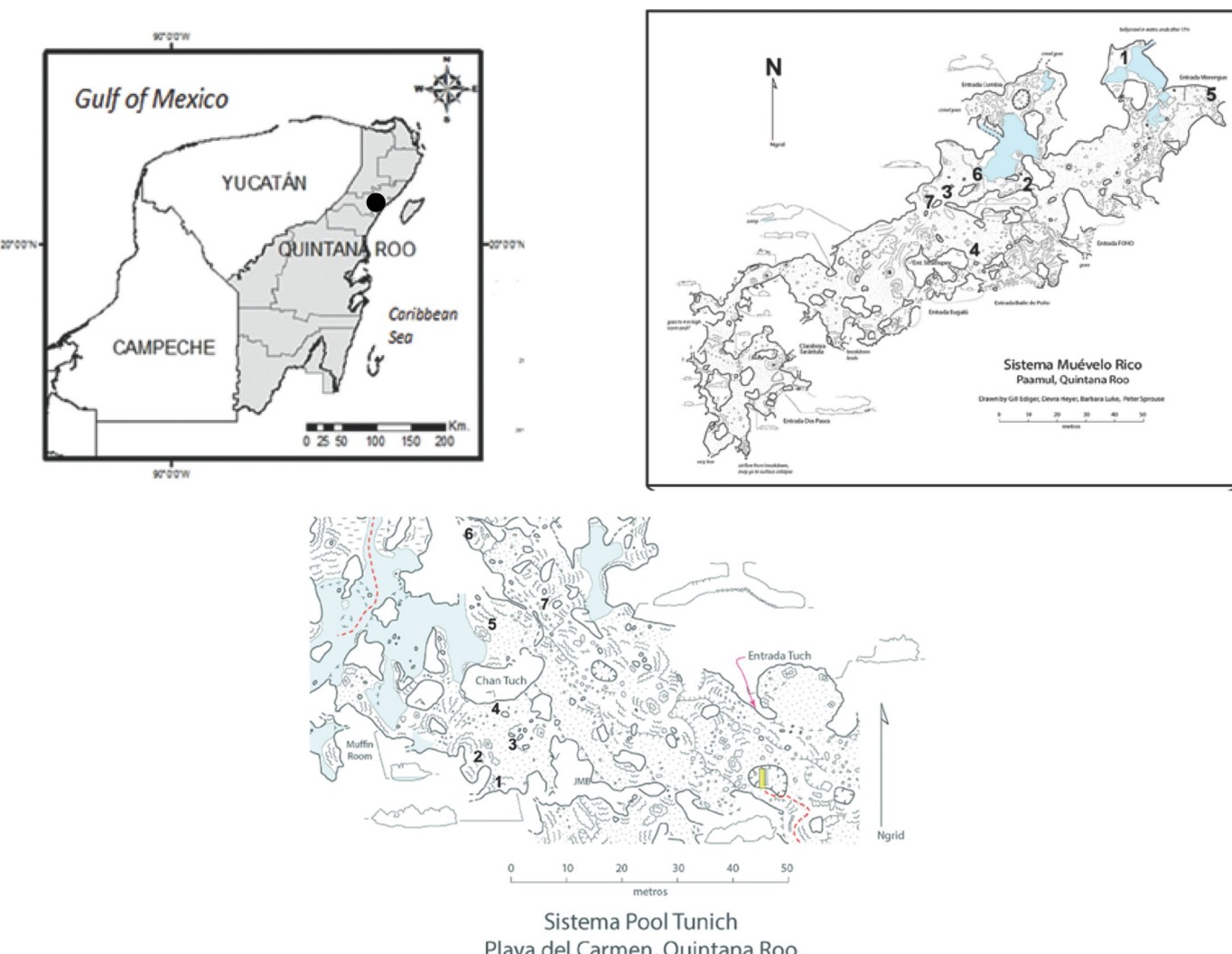

**Fig 1. Locator map for caves and sampling sites in the two caves.** Cave maps courtesy of Peter Sprouse. The open source base maps are from INEGI (Instituto Nacional de Estadistica y Geografia). From: Mejía-Ortíz et al. [15].

Taken together, the two caves represent two very different cave environments—relatively large with numerous surface connections (Sistema Muévelo Rico) to very large and less connected with the surface (Río Secreto [Tuch]).

## Relative humidity measurement

Relative humidity (along with temperature, see Mejía-Ortíz et al. [15,28]) was measured at hourly intervals for the following dates:

- Sistema Muévelo Rico—5 April 2015 to 28 March 2016, n = 8593

- Río Secreto (Tuch entrance)—25 September 2018 to 26 October 2019, n = 9515

Onset Computer Corporation HOBO[TM] U23 Pro v2 data loggers were used to measure relative humidity, and readings were accurate to ±5% for RH above 90%, with a resolution of 0.05%. However, we found that accuracy was better than reported. Nearby sensors and

temporally nearby measurements were consistent to more than 0.1% RH. Eight sensors were installed in Sistema Muévelo Rico, and one failed. Seven sensors were installed in Río Secreto, and all functioned for the entire measuring period.

### Faunal inventory

A preliminary faunal inventory was done in Río Secreto (Tuch) by employing a visual census for 30 person minutes at each station at the time the dataloggers were installed. Species were identified to morpho-species and species with reduced eyes and pigment (troglomorphs) were recorded. Previously analyzed data from Sistema Muévelo Rico [28] was used for comparative purposes.

### Data analysis

RH data from these two caves presented a number of statistical challenges. First, the data are percentages and bounded by 100. Second, the large majority of the values were 100, and for most stations, deviations from 100 were short-term, typically lasting less than 24 hours, and thus there was a clear baseline of 100 percent, populated with short-term deviations. We created two types of variables. First, in those cases where deviations were uncommon (all of the Río Secreto (Tuch) sites and sites 3,6, and 7 in Sistema Muévelo Rico), we tabulated the number of deviations of two hours or more from 100 percent for each month. Single hour deviations were not tabulated to reduce noisiness in the data and to eliminate very small deviations. Secondly, we created a daily variable for the remaining Sistema Muévelo Rico sites that is the proportion of hourly measurements in a day that had RH<100. For example, 0.2 means 20% of 24 measurements were less than 100% RH. General linear models (GLM) with non-constant variance and covariances among observations were used to estimate the mean seasonal RH proportions for each cave separately. The model included fixed effects of season within year and sensor; temporal autocorrelation of the observations was captured by assuming the residuals were correlated according to a autoregressive process with a lag of one (AR(1)); and variance was assumed to differ by season. The AR(1) covariance was chosen because temporal autocorrelations showed a strong value at a 1 day lag but a small partial autocorrelation value for a lag of 2 days (results not shown). It was expected that some seasons would have more variable values than others. Mean daily RH itself was analyzed in a similar way but did not meet the usual general linear model assumptions and yielded no significant results (not reported here).

Basic statistics (means, minima, maxima, and coefficients of variation) were calculated in EXCEL$^{TM}$, as were graphs of temporal patterns. Daily RH means were also generated in EXCEL$^{TM}$ for comparison with mean surface RH from Cozumel Air Force base, the closest monitoring point, which was approximately 20 km from the caves. Generalized linear models were calculated in SAS v9.4 (SAS Institute, Inc., Cary, NC).

Spectral analyses were done on hourly data to detect possible (daily) cycles. Cycles up to a period of 600 hours (25 days) were reported. Fisher's kappa tested for deviation from white noise. Analyses were done using JMP$^{®}$ Pro 13.2.0 (©2016 SAS Institute, Inc. Cary, NC).

## Results

### Overall patterns

The broad scale patterns of variation are summarized in Table 1. Mean RH at all sites was greater than 97 percent, even at the entrance of Sistema Muévelo Rico, and the lowest individual value was 69 percent, at the entrance to Sistema Muévelo Rico. Two sites in Río Secreto (Tuch) showed no variation, and RH was always 100 percent at these sites. For five of seven sites in Río Secreto (Tuch), mean RH was 100% and for the other two, the mean was greater than 99.9% (Table 1).

**Table 1. Overall summary of RH data for Río Secreto (Tuch) and Sistema Muévelo Rico.**

| Cave | Station | Lux | Mean RH | Min RH | Max RH | Percent non-saturation | CV RH | CV Temp | N troglo-morphic spp | Distance to entrance (m) |
|---|---|---|---|---|---|---|---|---|---|---|
| Sistema Muévelo Rico | 1 | <0.1 | 99.71 | 95.16 | 100 | 26.7 | 0.65 | 5.05 | 5 | 35 |
| | 2 | <0.1 | 98.24 | 87.95 | 100 | 57.9 | 2.46 | 5.78 | 5 | 20 |
| | 3 | <0.1 | 99.95 | 92.13 | 100 | 3.0 | 0.37 | 6.54 | 4 | 33 |
| | 4 | <0.1 | 98.19 | 77.13 | 100 | 45.3 | 3.06 | 8.49 | 1 | 9 |
| | 5 | 466 | 97.06 | 69.12 | 100 | 52.7 | 4.53 | 9.82 | 3 | 0 |
| | 6 | 0.2 | 99.98 | 95.39 | 100 | 2.0 | 0.19 | 5.14 | 4 | 16 |
| | 7 | <0.1 | 99.96 | 89.31 | 100 | 2.2 | 0.41 | 5.63 | 3 | 21 |
| Río Secreto (Tuch) | 1 | 0 | 99.99 | 90.97 | 100 | 0.4 | 0.18 | 3.70 | 4 | 17 |
| | 2 | 0 | 100 | 100 | 100 | 0.0 | 0.00 | 5.53 | 3 | 26 |
| | 3 | 0 | 100 | 100 | 100 | 0.0 | 0.00 | 5.13 | 2 | 8 |
| | 4 | 0 | 100 | 100 | 100 | 0.1 | 0.02 | 4.96 | 3 | 15 |
| | 5 | <0.1 | 100.00 | 99.02 | 100 | 0.3 | 0.03 | 4.73 | 4 | 5 |
| | 6 | 0.8 | 100.00 | 96.49 | 100 | 0.3 | 0.06 | 6.27 | 3 | 9 |
| | 7 | 7.7 | 99.96 | 89.34 | 100 | 2.0 | 0.41 | 5.14 | 1 | 3 |

T is temperature, RH is relative humidity, CV is the coefficient of variation, N is number.

Non-saturation is defined as RH<0.995.

Light intensity in lux and the coefficient of variation for temperature at the same stations are shown for comparison. Light and temperature data are from Mejía-Ortíz et al. [15,28].

The percentage of time of deviation from 100% RH was always less than 2% Río Secreto (Tuch) and only above 50% in two sites in Sistema Muévelo Rico, including one right at the entrance. Variability, as measured by the coefficient of variation was always less for RH than for temperature. While the temporal variation in light was not measured, there were four sites with invariant absence of light. RH was more variable in Sistema Muévelo Rico, and mean RH was correlated with distance to an entrance (Fig 2).

The time courses for RH in the caves are very different from those of surface RH at the nearby island of Cozumel (Fig 3). In the case of Río Secreto (Tuch) surface RH was always lower, and there was not overlap of the RH curves for the two sites, even when the most variable site in Rio Secreto (#7) was used for the comparison. In the case of Sistema Muévelo Rico, overlap, even in the case of the sensor placed right at an entrance (#5), RH overlapped only on a few days in May and June, the end of the dry season for this sampling year.

## Seasonal pattern of RH

The temporal pattern in Río Secreto (Tuch) was one of short downward spikes (Fig 4), with three sites being invariant with respect to RH. The deviations from 100% RH lasted between one and 49 hours, with most being only one or two hours in duration. Only one lasted more than 24 hours. The spikes are concentrated from September to January, in the end of the rainy season and start of the nortes season (Fig 5). Most of the spikes were in the afternoon, and in half the cases, the downward spike in RH was preceded by a slight rise in temperature. The one dry season dip occurred at 10PM when temperature was dropping. Overall, there is a weak seasonal pattern of constancy of RH outside the nortes season, where there are short downward spikes in RH. Mean RH does not vary; only the frequency of downward spikes varies.

The temporal pattern of RH in Sistema Muévelo Rico was more complicated (Fig 6). Three of the sites—3,6, and 7—show the Río Secreto pattern of brief downward spikes. However,

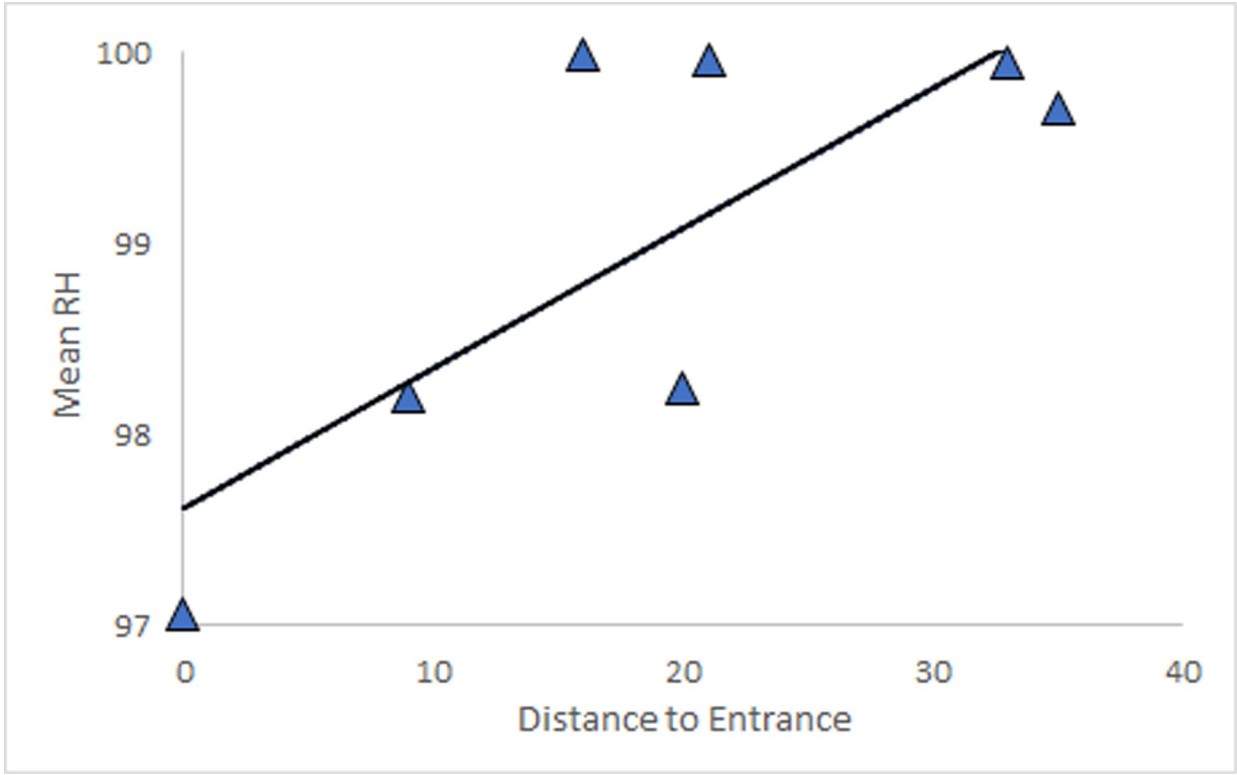

**Fig 2. Relationship between mean RH and distance from an entrance for Sistema Muévelo Rico.** The linear regression is significant (p = .041, $R^2 = 0.60$), with a slope of 0.07. Río Secreto (Tuch) showed almost no variation in mean RH at different stations (Table 1).

their monthly distribution is different. Like Río Secreto (Tuch), a number of spikes occur at the end of the Nortes season, but unlike Río Secreto (Tuch), there is a second peak at the end of the dry season (Fig 7).

At the other four sites (1,2,4, and 5), downward spikes occur but there are also extended periods where RH falls below 100 percent (Fig 6). Differences among these sites and among seasons (nortes, dry, and rainy) were analyzed. Site, season, and their interaction were all statistically significant (Table 2). According to the model, the rainy season had the lowest frequency of deviations from 100% RH and the dry season had the highest. The observed patterns, for the four individual sites are more complicated (Table 3), indicating the importance of site by season interactions (Table 2). In two sites (4 and 5), the nortes season showed the lowest mean proportion of deviations for 100% RH, and at site 1, the nortes season showed the highest mean proportion of deviations from 100% RH.

The differences in pattern cannot be explained by distance to the nearest entrance (Table 1). Those sites with the "Tuch" pattern of short term deviations from 100% RH are not the farthest from the entrance. Distance to an entrance only captures part of the extent of surface environmental influence, and size and aspect are also important, especially in Sistema Muévelo Rico, with multiple entrances.

## Daily pattern of RH

In the case of daily cycles in Río Secreto (Tuch), only sites 1,6, and 7 were variable enough to analyze for daily cycles. Although Fisher's kappa test indicated the pattern was different from

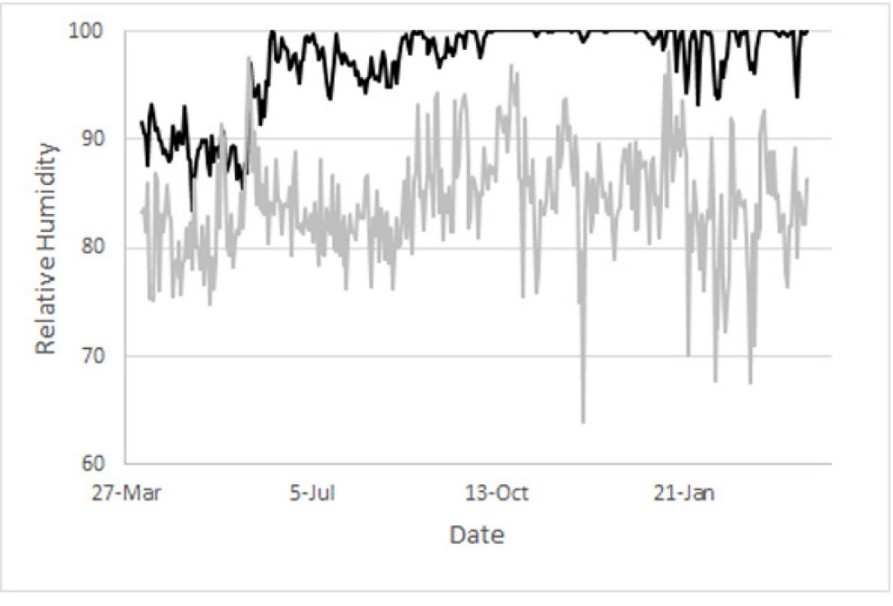

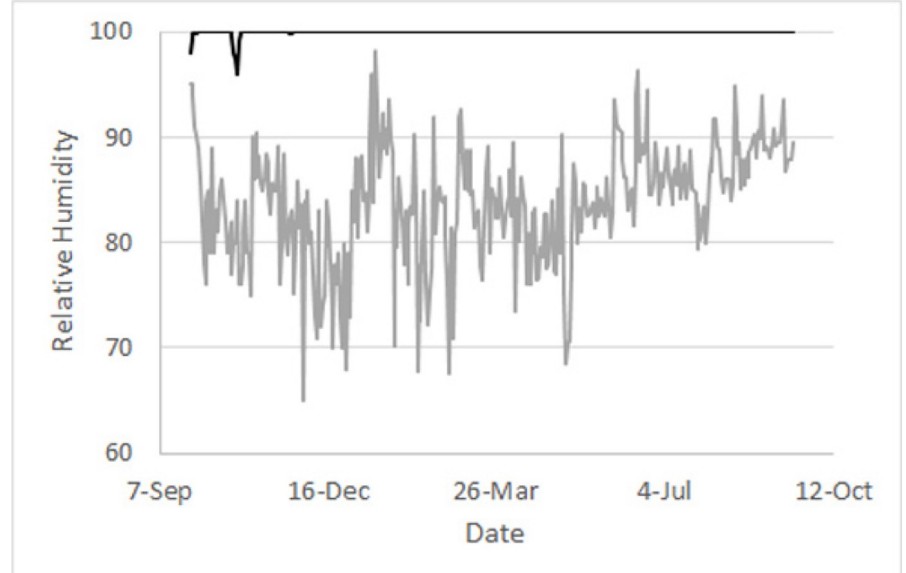

**Fig 3. Comparison of cave and surface relative humidity.** Top: Daily average RH (black line) for Río Secreto (Tuch) site 7 (the most variable site) and daily average RH (gray line) for Cozumel for the period from 25 September 2018 to 26 October 2019. Bottom: Daily average RH (black line) for Sistema Muévelo Rico site 5 (the most variable site) for the period from 5 April 2015 to 28 March 2016. Cozumel data courtesy of Sub-lieutenant Jhosep Guadarrama Espinoza of Mexican Air Force stationed in Cozumel.

white noise, none of the three sites had a clear peak at 24 hours (Fig 8), with the possible exception of site 7.

For Sistema Muévelo Rico, the results are shown in Fig 9. As expected, the entrance station (#5) showed a very strong 24 hour signal, as did stations 2 and 4. Stations 1 and 3 showed a relatively clear signal, but it was very weak. This is not surprising since RH in the cave was invariant for about half of the year (Fig 7). Stations 6 and 7 showed an even weaker signal, and they were also invariant for much of the year.

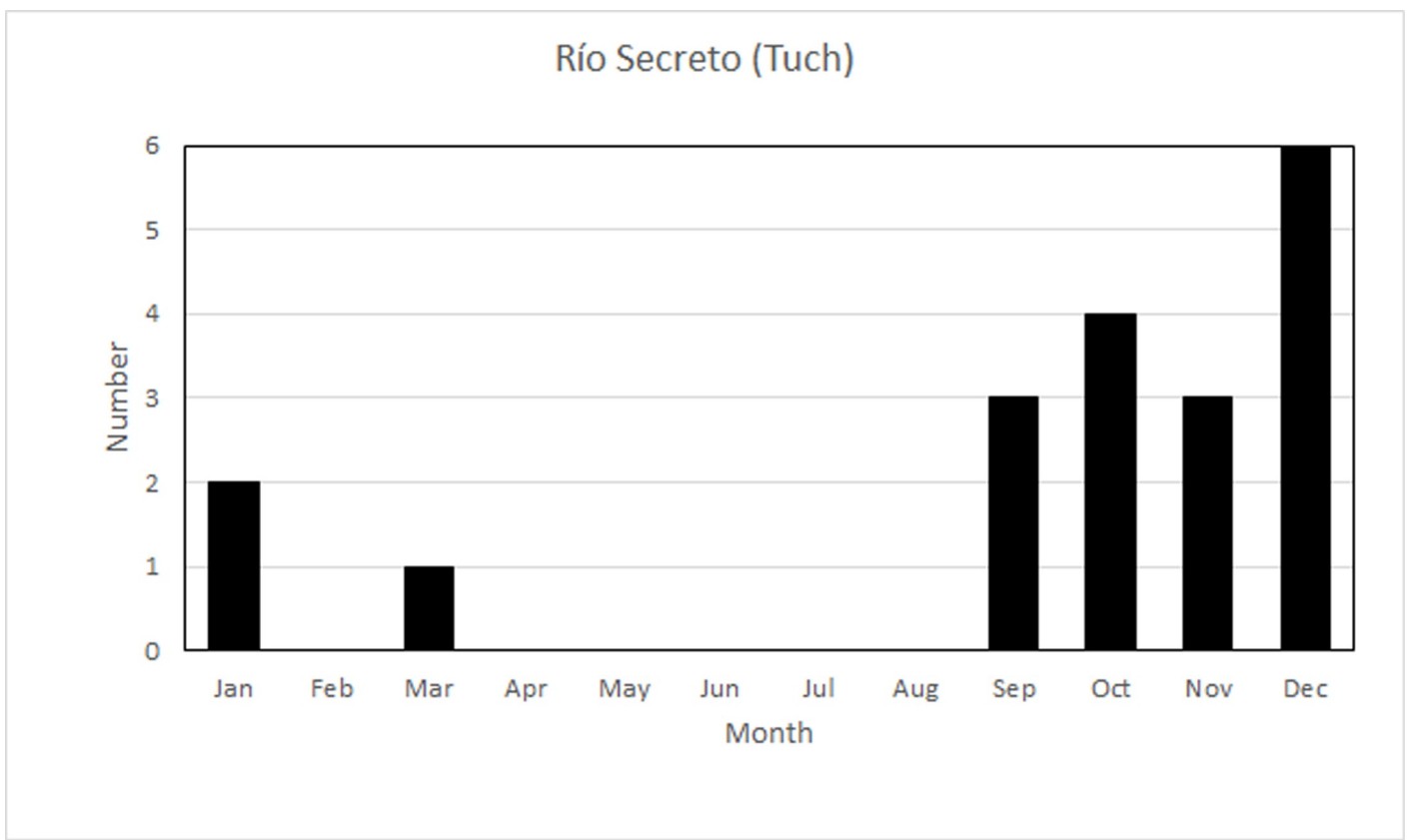

**Fig 4. Variation in RH at the seven sites in Río Secreto (Tuch).** Site 7 is closest to the entrance (see Table 1).

### Fauna-RH connections

The number of troglomorphic species, by station, is shown in Table 1. The data for Sistema Muévelo Rico are the result of four censuses [28] and the data for Río Secreto (Tuch) are the result of a single census, so comparisons between the caves are not possible. In Río Secreto (Tuch), invariant or nearly invariant sites had no more species than other sites. In Sistema Muévelo Rico likewise, variable RH sites had no fewer species than invariant sites. The number of troglomorphic species was not significantly correlated with variation in RH for either cave, and in fact the correlations were negative, rather than positive. Sample size was very low (n = 5) so little can be inferred from this.

## Discussion

### Patterns and variability of RH

Relative humidity was less variable than temperature at all stations with consistently lower coefficients of variation (Table 1). For seven sites in Río Secreto (Tuch), three sites showed no variation in RH and the other four sites had coefficients of variation of less than 1 percent. In Sistema Muévelo Rico, four sites had coefficients of variation of less than 1 percent. No site in either cave had a temperature coefficient of variation of less than 3.7 percent (site 1 in Río Secreto (Tuch)).

While it was not possible to measure light more than once, the only aphotic sites were four in Río Secreto (Tuch). The daily light-dark cycle at other sites may result in coefficients of

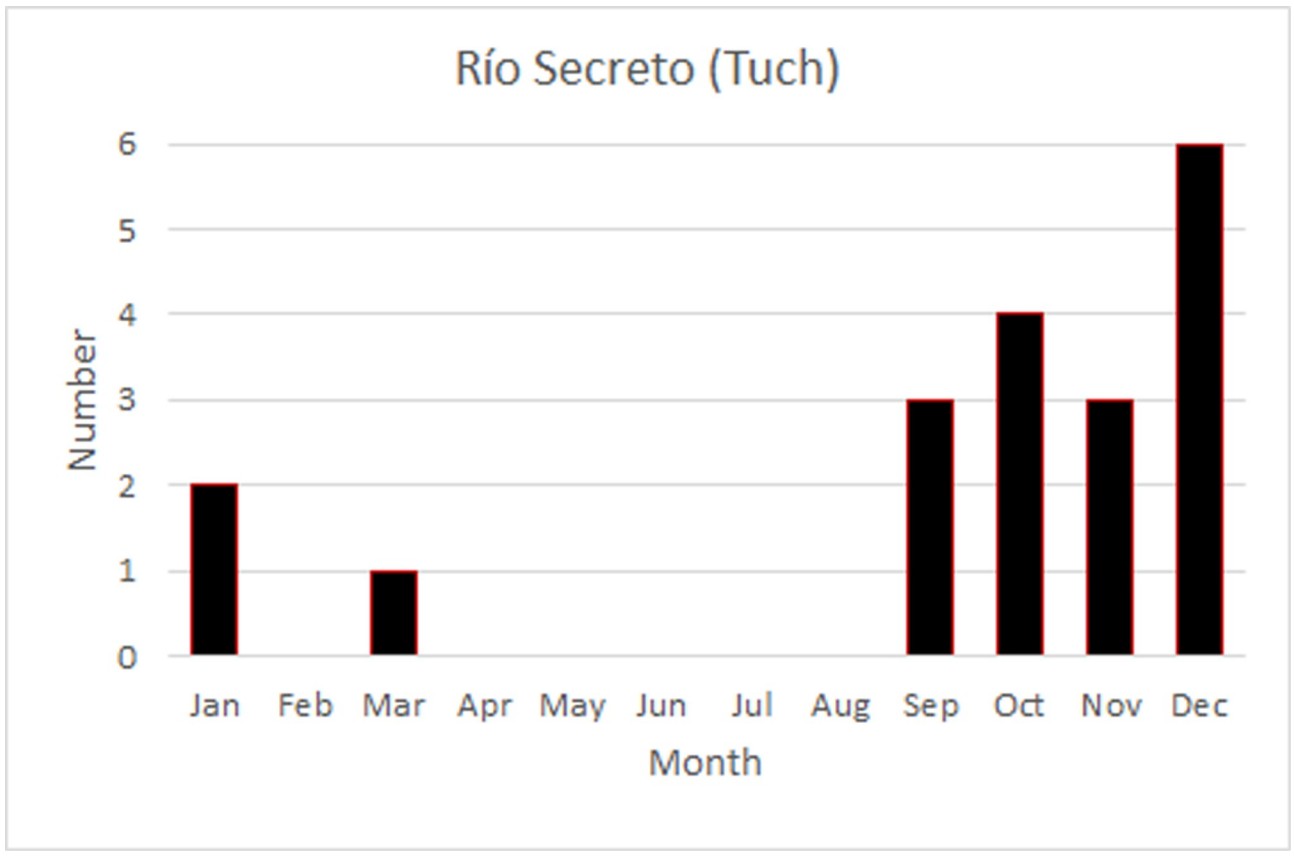

**Fig 5. Distribution of downward spikes in RH, by month, for sites in Río Secreto (Tuch).**

variation greater than 1 percent. In addition to the daily light cycle, there is also a seasonal effect. Both the day length and apogee of the sun vary seasonally. Day length ranges from 10 hr 51 min to 13 hr 25 min [29]. All of this suggests that RH is less variable in these two caves than light, as well as temperature.

## Seasonal and daily cycles of RH

Seasonality of RH is present in both caves. In Río Secreto (Tuch), it is the clustering of deviations from 100% RH at the end of the nortes season and the start of the rainy season. In Sistema Muévelo Rico not only are there downward spikes, but there are also extended periods of RH below 100%. However, these periods are neither synchronous within the cave (Table 4) or with Río Secreto (Tuch) (Figs 4 and 6). This is in sharp contrast with the situation with temperature, where there was a clear, synchronous seasonality [15]. Why there is more variability in RH at a small spatial scale is unclear.

The daily pattern was one of diminished 24 hour periodicity compared to temperature. In Sistema Muévelo Rico sites, a daily cycle could be detected at all sites, albeit very weak in some. In Río Secreto (Tuch), a daily cycle was detectable in only one site.

## Is there a winter effect?

Numerous investigators have pointed out that in temperate caves there is a "winter" effect, with a reduction in relative humidity, largely the result of air movements [3,11,16,21,30].

**Sistema Muévelo Rico 3,6,7**

**Fig 6. Variation in RH at the seven sites in Sistema Muévelo Rico.**

According to Barr and Kuehne [31], these winter winds resulted in the absence of cave fauna in affected passages in Mammoth Cave, Kentucky. They only found animals in passages with RH above 94 percent, while some passages had RH near 80 percent. Barr and Kuehne [31] used sling psychrometer to measure RH and thus could only find RH away from walls, floors,

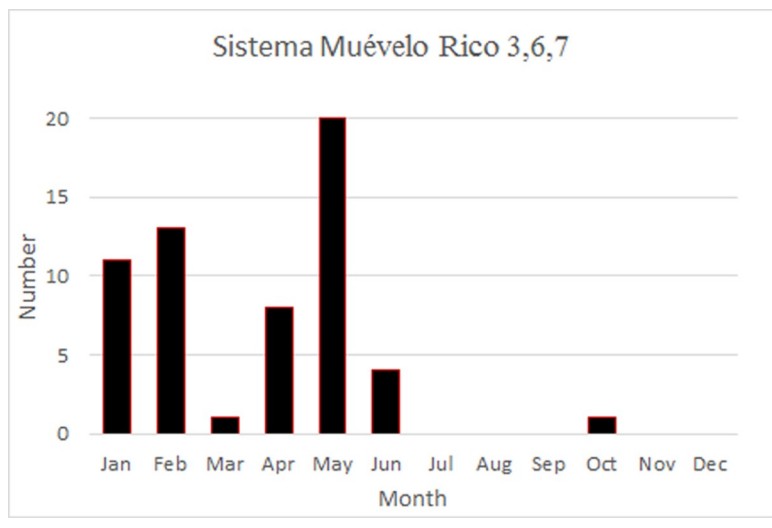

**Fig 7. Distribution of downward spikes in RH, by month, for sites 3,6, and 7 in Sistema Muévelo Rico.**

Table 2. Type III test of fixed effects in linear generalized model for proportion of hours not at 100% RH for four sites (1,2,4,5) in Sistema Muévelo Rico.

| Effect | Num DF | Den DF | F value | Pr>F |
|---|---|---|---|---|
| Site | 3 | 38.08 | 3.20 | 0.034 |
| Season | 2 | 32.4 | 3.32 | 0.049 |
| Site*Season | 6 | 34.33 | 4.41 | 0.0021 |

etc. In these sites, winds would be higher than along the substrate, and hence RH would be lower. Howarth [8] argued that terrestrial cave limited invertebrates have evolved to survive in 100 percent RH, a water saturated environment, by cuticular thinning [10], which allows for greater water exchange. The cost of this adaptation is water loss (and increased mortality) in non-saturated environments. Howarth [8] points out that water saturated environments are in some ways aquatic habitats.

It is not at all clear that there is any winter effect in Río Secreto, and that it appears diminished in Sistema Muévelo Rico. In Río Secreto, drops in RH, if they occur at all, are of very short duration (Figs 3 and 4). Most of the short duration downward spikes occur in the nortes season. In Sistema Muévelo Rico, there are periods of RH that are below saturation for extended periods of time (Fig 6), but it is unknown if the magnitude of the drops is sufficient to cause any physiological response from the organisms inhabiting the cave. No discernible effect in faunal composition through the seasons was found [28]. Perhaps the seasonal difference for winter is less in tropical surface habitats and this is the reason we found little evidence for it in caves.

Furthermore, it is not at all clear how general the winter effect in temperate zone caves is, and we know of no well documented RH measurements in a cave throughout a year that show it. Tobin et al. [32] report near constant RH in a small California marble cave but they did not monitor the cave from January through April. Several authors [30,31,33] report both on RH variation and winds in Mammoth Cave, but there are little quantitative data taken at regular intervals.

There are desert caves that are noticeably drier, but even in these cases, RH is rarely less than 80 percent, even in dusty passages. Probably the best studied is Torgac Cave in New Mexico [34]. The cave, developed in dolomite, is covered with gypsum minerals, which are formed as a result of evaporation [35,36]. RH in the cave in January ranged from 85 to 95 percent, both on the basis of sling psychrometer and electronic sensor readings [4].

Table 3. Observed mean daily proportion of hours RH<100 by site and season for Sistema Muévelo Rico.

| Site | Season | N Obs | Mean | Standard Error |
|---|---|---|---|---|
| 1 | Dry | 85 | 0.12 | 0.03 |
| | Nortes | 121 | 0.69 | 0.04 |
| | Rainy | 153 | 0.01 | 0.004 |
| 2 | Dry | 85 | 0.98 | 0.01 |
| | Nortes | 121 | 0.68 | 0.04 |
| | Rainy | 153 | 0.28 | 0.03 |
| 4 | Dry | 85 | 0.69 | 0.05 |
| | Nortes | 121 | 0.18 | 0.03 |
| | Rainy | 153 | 0.53 | 0.03 |
| 5 | Dry | 85 | 0.72 | 0.05 |
| | Nortes | 121 | 0.23 | 0.03 |
| | Rainy | 153 | 0.65 | 0.03 |

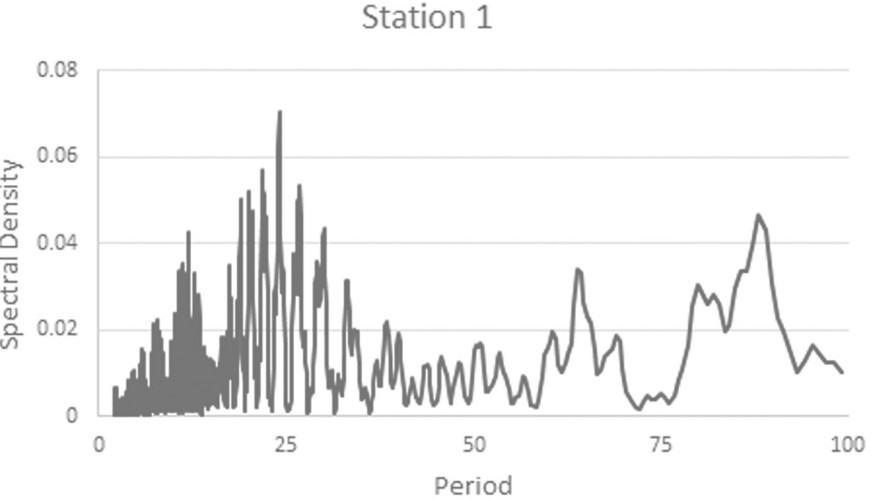

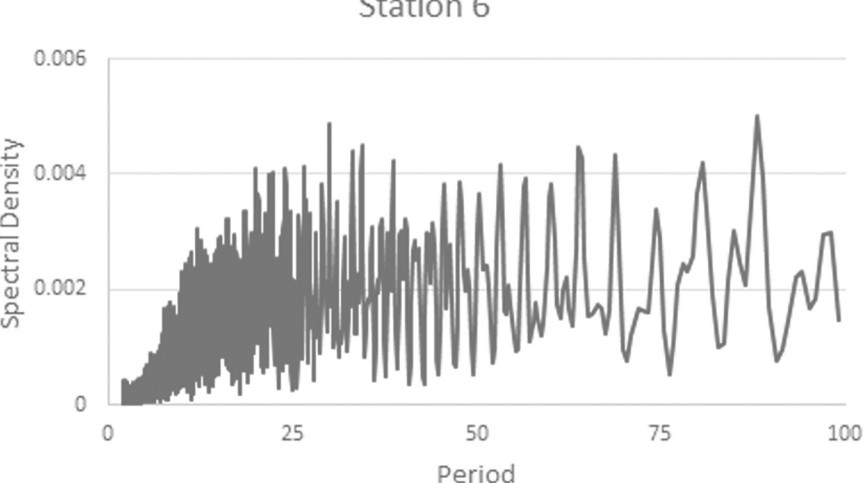

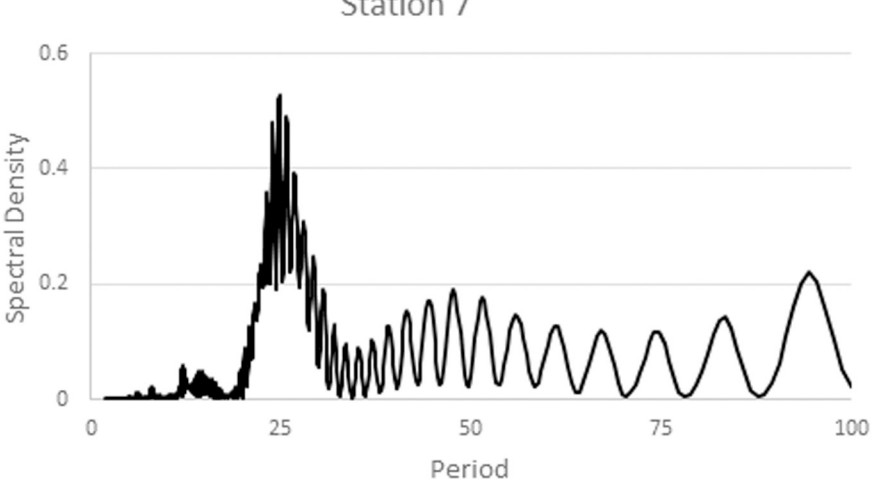

**Fig 8. Spectral analysis of sites 1,6, and 7 in Río Secreto (Tuch).** All sites were significantly different from white noise, according to Fisher's kappa test.

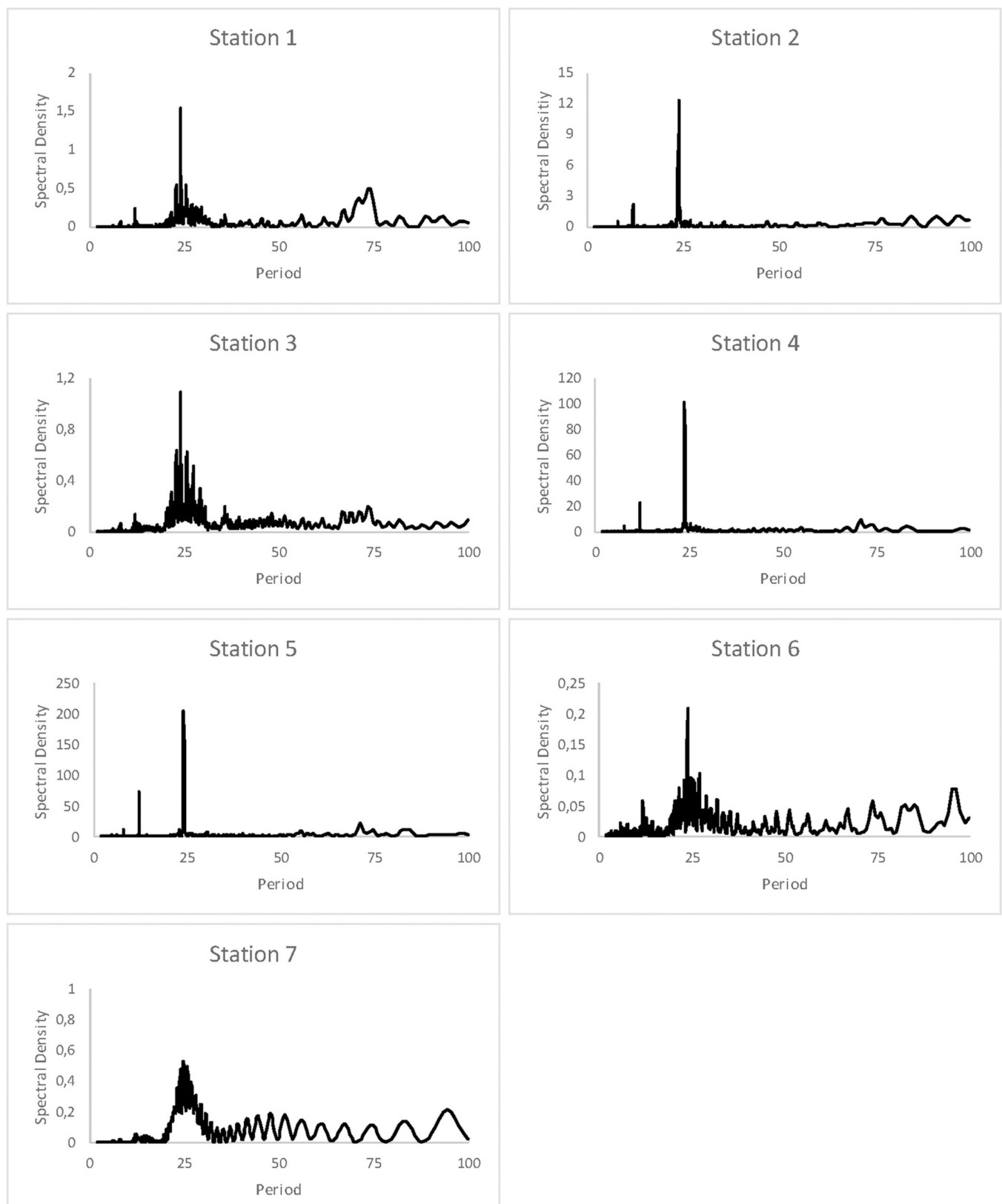

**Fig 9. Spectral analysis of cycles in RH at the seven monitoring stations in Sistema Muévelo Rico.** Spectral density indicates the strength of the signal. Strong cyclicity is seen at 24 hours. Reproduced with the permission of the authors of [28].

**Table 4. Presence/Absence of variation in temperature, light, and relative humidity in the two study caves, by station.**

| Cave | Station | Daily Cycle | | | Seasonality | | |
|---|---|---|---|---|---|---|---|
| | | Light | Temp. | RH | Light | Temp. | RH |
| Sistema Muévelo Rico | 1 | | | | | | X |
| | 2 | | | | | | X |
| | 3 | | | | | | ? |
| | 4 | | | | | | X |
| | 5 | | | | | | ? |
| | 6 | | | ? | | | X |
| | 7 | | | ? | | | ? |
| Río Secreto (Tuch) | 1 | X | ? | X | X | | X |
| | 2 | X | X | X | X | | X |
| | 3 | X | ? | X | X | | X |
| | 4 | X | ? | X | X | | X |
| | 5 | | ? | X | | | X |
| | 6 | | | ? | | | X |
| | 7 | | | ? | | | ? |

X's indicate an aphotic station, acyclic temperature, and acyclic RH. Question marks are cases where cyclicity is very weak.

Unfortunately, no seasonal data are available to know if RH is higher in the summer. The winter effect is understudied in general and especially so in tropical caves.

Rather than a winter effect there may be a hurricane period effect, because during hurricanes the humidity is close to 100% outside, and the rain has several effects on the energy sources, growth of roots, organic matter entrance, and of course on the changes in the temperature and humidity. This hurricane period is likely to change under global climate change, and this will likely have an important impact on the RH cycle.

## Comparison of patterns of RH, temperature, and light

Typically, caves are divided into three zones [2,3,31,32]: (1) an entrance zone with light, (2) an intermediate aphotic zone with variable temperature, and (3) a deep zone without light and constant temperature. How relative humidity fits into this scheme is not clear. In Río Secreto (Tuch), relative humidity was constant, or nearly so, not only in the dark zone, but also at two stations with light present (4 and 5, Table 2). In Sistema Muévelo Rico, which was chosen because of its extensive twilight zone and multiple entrances [28], only site 7 did not show a clear seasonal pattern (Fig 6), but the daily cycle was largely absent (Fig 7).

This pattern stands in sharp contrast with temperature. In Sistema Muévelo Rico, there was no zone of constant temperature and almost no zone of constant darkness (Table 4). Even in Río Secreto (Tuch), there was always a seasonal and daily cycle of temperature. It may well be that a constant temperature zone can be found deeper in the cave, the demonstration of a constant temperature zone in any cave remains elusive. Overall, 100 percent RH is more common than complete darkness, at least in the caves we studied.

## How biologically and geologically important is RH in caves?

RH is critically important in several geological processes. One of these involves carbonate dissolution—condensation corrosion, the condensation of warm, humid air to cold rock walls [19,23], which can be an important factor in speleogenesis in some circumstances [35]. As Badino [23] pointed out, cave atmospheres can be super-saturated, hence the possibility of

cloud formation. It is these conditions under which condensation corrosoion is important. Mineral precipitation in caves, especially gypsum minerals, is the result of evaporation [36]. Gypsum minerals can appear and disappear seasonally as RH in cave passages changes.

Howarth [8] proposed that because of the high humidity of caves that terrestrial species adapted by cuticular thinning which allowed for greater water movement across the integument. This morphological difference was demonstrated in the case of lycosid spiders living in lava tubes in Hawaii [10] and for a terrestrial isopod, *Titanethes alba*, in Slovenian caves [37]. Interestingly, this species is amphibious and can move in and out of water. Humphreys and Collis [38] showed that cave arthropods from the Cape Range of Australia showed greater water loss than epigean species presumably as the result of cuticular thinning. Cuticular thinning may well be a very common convergent trait, and is certainly worthy of further study.

As with the surface biota, climate change poses a grave threat to the subterranean fauna [39,40]. The terrestrial cave fauna is very sensitive both to the level of relative humidity [8] and its pattern [9]. Given the widespread occurrence of dripping water in caves, many (but not all) will equilibrate back to moisture saturation following a temperature increase, but the lag time for this to happen may be considerable, certainly decades and maybe longer [13,19], and thus impose a significant risk. In addition, if the tropical seasons, especially the hurricane season, changes, this will profoundly impact RH, and if stable RH is generally required (see [9]), then this too will have a major deleterious impact on the the terrestrial fauna.

## Conclusions

Compared to temperature, RH in the two caves studied had a very different temporal pattern, in spite of the fact that RH is strongly dependent on temperature. More than half of the sites showed constant RH or near constancy with brief dips, the constancy the result of the presence of some standing water in both caves. These deviations tended to occur at the end of the nortes and beginning o In the other sites, those closer to the entrance, there was a seasonal difference in RH, but the details differed between the two caves. Some of these sites, especially in Sistema Muévelo Rico, also showed a 24 hour cycle, but these cycles were not as prominent as those for temperature. While RH is known to be important to the physiology of the cave fauna, troglomorphic species were more or less uniformly spread within each cave, with no concentration in any particular RH regime.

## Supporting information

**S1 Table. Hourly relative humidity data for Sistem Muévelo Rico.**
(XLSX)

**S2 Table. Hourly relative humidity data for Río Secreto (Tuch).**
(XLSX)

## Acknowledgments

Alberto Rivero gave permission to visit the Sistema Muévelo Rico. Tania Ramirez of Río Secreto Natural Preserve gave permission to visit the Tuch entrance of Río Secreto and provided logistical support. We thank Sub-lieutenant Jhosep Guadarrama Espinoza from Mexican Air Force for providing the meteorological data for Cozumel. Stefano Mammola provided a number of useful suggestions that greatly improved the manuscript. No permits were needed since no organisms were collected for this study.

## Author Contributions

**Conceptualization:** Luis Mejía-Ortíz, Mary C. Christman, Tanja Pipan, David C. Culver.

**Data curation:** Mary C. Christman, Tanja Pipan.

**Formal analysis:** Mary C. Christman.

**Funding acquisition:** Luis Mejía-Ortíz.

**Investigation:** Luis Mejía-Ortíz, Tanja Pipan, David C. Culver.

**Methodology:** Tanja Pipan.

**Writing – original draft:** David C. Culver.

**Writing – review & editing:** Luis Mejía-Ortíz, Mary C. Christman, Tanja Pipan.

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
