## [Decision Letter · Decision Letter 0]

2 Aug 2021

PONE-D-21-10979

What's the relative humidity in tropical caves?

PLOS ONE

Dear Dr. Culver,

Thank you for submitting your manuscript to PLOS ONE. After careful consideration, we feel that it has merit but does not fully meet PLOS ONE’s publication criteria as it currently stands. Therefore, we invite you to submit a revised version of the manuscript that addresses the points raised during the review process. Reviewer's comments are appended below. 

We look forward to receiving your revised manuscript.

Kind regards,

Mahendra Singh Dhaka, Ph.D.

Academic Editor

PLOS ONE

Additional Editor Comments (if provided):

The authors need to strengthen the language and scientific interpretations of the facts throughout the manuscript. Also, the conclusion part should be strong based upon the findings. Recent references should also be included.

Journal Requirements:

Reviewers' comments:

Reviewer's Responses to Questions

**Comments to the Author**

1. Is the manuscript technically sound, and do the data support the conclusions?

Reviewer #1: Yes

2. Has the statistical analysis been performed appropriately and rigorously? 

Reviewer #1: Yes

3. Have the authors made all data underlying the findings in their manuscript fully available?

Reviewer #1: Yes

4. Is the manuscript presented in an intelligible fashion and written in standard English?

Reviewer #1: Yes

5. Review Comments to the Author

Reviewer #1: A well-written and original study about the physical behavior of relative humidity (RH) in the atmosphere of two tropical caves. The results are contextualized in an ecological perspective, namely how the subterranean fauna may be influenced by variations in the RH. I truly enjoyed reading the study and learned a lot from it. I have only a couple of suggestions and a few requests for clarification. I hope this revision will be useful.

------- MAIN SUGGESTIONS:

1) Badino was pursuing an interesting line of research on the supersaturation of RH in caves. He told me about this apparent paradox: in a cave, over short periods, the relative humidity can be above 100%. The thing is that, in some caves, you may have a situation where you reach 100% RH, water condensate on all surfaces (that is, everything is covered by a film of water), but the air is so pure that you may reach a point where there are no more air impurities particles (condensation nuclei) where water can condensate. Thus, this “exceeding” water remains stuck in the atmosphere, which can exceed 100% (up to values of 101–102%). The available dataloggers cannot detect this. Before he died, Badino was working closely with engineers to design a tool to measure the “hidden” water content in super-saturated cave atmospheres.

I’m not sure if you want to mention this (e.g. L108–110). Unfortunately, I don’t remember the technicalities and I don’t have the physical understanding to better explain this. The topic was only briefly discussed in this publication (section “Condensation nuclei”):

Badino G (2004) Clouds in caves. S. E. K. A. 2:1–8 (https://www.researchgate.net/publication/26448070_Clouds_in_caves)

It is an obscure publication in a conference proceeding; If you cannot find it just e-mail me and I will share it with you. I think it could be a relevant publication to other sections of the paper as well.

2) I think it could be worth mentioning future climate change when you discussed the sensitivity of cave organisms to RH variations. The fact is that RH and temperature are linked by the Clapeyron equation, e.g.

Badino G., 2004 - Cave temperature and Global Climate change. International Journal of Speleology, 33 (1/4): 103-114.

If subterranean climate will change, there will be the cascading effects on RH, which potentially will render some suitable habitats unsuitable. We discussed this here:

Mammola, S. et al. (2019). Climate change going deep: the effects of global climatic alterations on cave ecosystems. The Anthropocene Review, 6(1-2), 98-116.

Also, changing precipitation regimes and surface desertification due to climate change will likely feedback to affect the availability of water in caves, with direct repercussions on relative humidity patterns. This is briefly discussed here:

Sánchez-Fernández, D. et al. (2021). Don’t forget subterranean ecosystems in climate change agendas. Nature Climate Change, 1-2.

I think a quick mention of this conservation issue could increase the generality of the study (e.g. in the section at L421). That said, I will understand if you will decide to ignore this: being one of my main lines of research, I have a bias towards overemphasizing the importance of climate change underground.

------- ADDITIONAL MINOR LINE COMMENTS:

-L 111: This is a more recent one relating RH and cave species abundance:

Nicolosi, G. et al. (2021). Microhabitat selection of a Sicilian subterranean woodlouse and its implications for cave management. International Journal of Speleology.

- L 195: The structure of the model is not clear to me. What was the random structure in the model? Season? Did you fit two GLMM for the two caves, or everything together with cave identity as a fixed factor?

- L196: “ … of ar(1) ...” the meaning of this expression is not clear to me.

- Table 1: please define acronyms in the caption.

- L319: the relationship between RH and Fauna: did you tested this statistically?

- L362: Winter effect. Could it partly be because external seasonal variations in tropical climates are more smooth?

-L 392: very interesting the hurricane period effect–another factor that may be affected by global CC.

---

## [Author Response · Author response to Decision Letter 0]

10 Aug 2021

With respect to the Reviewer’s comments (please note the line numbering seems a bit haywire in the track changes version with some gaps in numbering between pages):

1. We have added discussion of climate change at the end of the discussion as well as in the introduction. We mention the linkage of temperature and RH via the Clapeyron equation in line 102. 

2. With respect to Badino’s work on supersaturated atmospheres in caves (clouds in caves), we discuss this in connection with condensation corrosion (line 595), and mention it elsewhere in the manuscript.

3. The description of GLMM (line 265) was rewritten and expanded, and ar(1) is explained.

4. The acronyms in Table 1 are defined at the bottom of the Table.

5. The relationship between RH and fauna (or more correctly, the lack thereof) is discussion beginning on line 438. 

6. We include the possibility that the winter effect is minimal because of reduced external seasonal variation in tropical climate on line 519.

7. We include a mention of the hurricane effect on line 555

8. We have added five additional recent references on global warming in caves.

Please let me know if more information is required.

Sincerely,

David C Culver

---

## [Decision Letter · Decision Letter 1]

31 Aug 2021

PONE-D-21-10979R1

What's the relative humidity in tropical caves?

PLOS ONE

Dear Dr. Culver,

Thank you for submitting your manuscript to PLOS ONE. After careful consideration, we feel that it has merit but does not fully meet PLOS ONE’s publication criteria as it currently stands. Therefore, we invite you to submit a revised version of the manuscript that addresses the points raised during the review process. Editor's comments appended below. 

We look forward to receiving your revised manuscript.

Kind regards,

Mahendra Singh Dhaka, Ph.D.

Academic Editor

PLOS ONE

Journal Requirements:

Additional Editor Comments:

Based upon the study, the authors need to incorporate 'Conclusion', before supporting information and acknowledgements.

Reviewers' comments:

Reviewer's Responses to Questions

**Comments to the Author**

1. If the authors have adequately addressed your comments raised in a previous round of review and you feel that this manuscript is now acceptable for publication, you may indicate that here to bypass the “Comments to the Author” section, enter your conflict of interest statement in the “Confidential to Editor” section, and submit your "Accept" recommendation.

Reviewer #1: All comments have been addressed

2. Is the manuscript technically sound, and do the data support the conclusions?

Reviewer #1: Yes

3. Has the statistical analysis been performed appropriately and rigorously? 

Reviewer #1: Yes

4. Have the authors made all data underlying the findings in their manuscript fully available?

Reviewer #1: Yes

5. Is the manuscript presented in an intelligible fashion and written in standard English?

Reviewer #1: Yes

6. Review Comments to the Author

Reviewer #1: (No Response)

7. PLOS authors have the option to publish the peer review history of their article (what does this mean?). If published, this will include your full peer review and any attached files.

Reviewer #1: **Yes: **Stefano Mammola

---

## [Author Response · Author response to Decision Letter 1]

1 Sep 2021

Conclusion added

No other comments were received.

---

## [Editor Report · Decision Letter 2]

6 Sep 2021

What's the relative humidity in tropical caves?

PONE-D-21-10979R2

Dear Dr. Culver,

We’re pleased to inform you that your manuscript has been judged scientifically suitable for publication and will be formally accepted for publication once it meets all outstanding technical requirements.

Kind regards,

Mahendra Singh Dhaka, Ph.D.

Academic Editor

PLOS ONE
---

## [Editor Report · Acceptance letter]

13 Sep 2021

PONE-D-21-10979R2 

What’s the Relative Humidity in Tropical Caves? 

Dear Dr. Culver:

I'm pleased to inform you that your manuscript has been deemed suitable for publication in PLOS ONE. Congratulations! Your manuscript is now with our production department. 

Kind regards, 

on behalf of

Dr. Mahendra Singh Dhaka 

Academic Editor

PLOS ONE